# Different Responses to Stress, Health Practices, and Self-Care during COVID-19 Lockdown: A Stratified Analysis

**DOI:** 10.3390/ijerph18052253

**Published:** 2021-02-25

**Authors:** Elena Bermejo-Martins, Elkin O. Luis, Ainize Sarrionandia, Martín Martínez, María Sol Garcés, Edwin Y. Oliveros, Cristian Cortés-Rivera, Maider Belintxon, Pablo Fernández-Berrocal

**Affiliations:** 1School of Nursing, University of Navarra, 31009 Pamplona, Spain; ebermejo@unav.es (E.B.-M.); mbelintxon@unav.es (M.B.); 2Navarra Institute for Health Research, IdiSNA, 31009 Pamplona, Spain; mmvillar@unav.es; 3Psychological Processes in Education and Health Group, School of Education and Psychology, University of Navarra, 31009 Pamplona, Spain; 4Cognitive and Affective Methods in Psychology CAMP, School of Education and Psychology, University of Navarra, 31009 Pamplona, Spain; 5Faculty of Psychology, University of the Basque Country, 20018 Donostia-San Sebastián, Spain; ainize.sarrionandia@ehu.eus; 6Colegio de Ciencias Sociales y Humanidades, Universidad San Francisco de Quito USFQ, Instituto de Neurociencias, Quito 170901, Ecuador; sgarces@usfq.edu.ec; 7Faculty of Psychology, University of San Buenaventura, Bogota 1008, Colombia; eoliveros@usbbog.edu.co; 8Faculty of Psychology, Universidad del Desarrollo, Santiago 7590943, Chile; crcortesr@udd.cl; 9Faculty of Psychology, University of Málaga, 29071 Malaga, Spain; berrocal@uma.es

**Keywords:** COVID-19, health practices, stress, self-care, cross-cultural study

## Abstract

The aim of the present cross-sectional study was to analyze the differential impact of the first COVID-19 lockdown (3 April 2020) on stress, health practices, and self-care activities across different Hispanic countries, age range, and gender groups. One thousand and eighty-two participants from Spain, Chile, Colombia, and Ecuador took part in this study. Irrespective of the country, and controlling for income level, young people, especially females, suffered a greater level of stress, perceived the situation as more severe, showed less adherence to health guidelines, and reported lower levels of health consciousness, in comparison to their male peers and older groups. However, in the case of self-care, it seems that older and female groups are generally more involved in self-care activities and adopt more healthy daily routines. These results are mostly similar between Colombia, Ecuador, and Spain. However, Chile showed some different tendencies, as males reported higher levels of healthy daily routines and better adherence to health guidelines compared to females and people over the age of 60. Differences between countries, genders, and age ranges should be considered in order to improve health recommendations and adherence to guidelines. Moreover, developing community action and intersectoral strategies with a gender-based approach could help to reduce health inequalities and increase the success of people’s adherence to health guidelines and self-care-promoting interventions. Future studies should be addressed to explore the possible causations of such differences in more cultural-distant samples and at later stages of the current outbreak.

## 1. Introduction

As of 14 April 2020, more than 18,056 people have died as a result of the coronavirus disease 2019 (COVID-19) in Spain; more than 127 in Colombia; more than 92 in Chile; and more than 369 in Ecuador [1]. This led to unprecedented efforts to institute the practice of “social distancing” in most countries around the world, resulting in a strict national lockdown and affecting the population’s usual functioning and daily lives. Although these measures are crucial to mitigating the spread of this disease, they are undoubtedly affecting people’s health and well-being in the short and long term. As a consequence, people from different countries have suffered (and continue to suffer) several psychological symptoms and health problems [2,3,4,5,6].

In contrast, this situation can offer the opportunity to progress in terms of health promotion and prevention strategies [7]. In this sense, from a salutogenic perspective, it is important to move beyond individual risk factors such as tobacco use, to social and structural forces on health, and to “salutary” factors such as education or people’s capability to remain healthy [8]. The salutogenic approach highlights the importance of promoting health assets and the active role of people in creating health. Given that health arises from the interplay between people and their context, it must be considered that people have a critical role in bringing about change [9]. According to Morgan and Ziglio [10], these health assets represent any factor or resource that enhances the ability of individuals, communities, and populations to maintain and sustain health and well-being and to help to reduce health inequalities. Therefore, these assets can operate at the level of the individual, family or community, and population as protective and promoting factors to buffer against life’s stressors. 

From a health-assets approach, promoting self-care activities could be key to bolstering physical and mental health at the individual level [11]. In fact, during the current pandemic, it has been widely advised to engage in self-care activities to reduce stress, along with maintaining a healthy lifestyle as a protective factor against the virus complications [12,13]. In the same vein, people who suffered a higher impact of quarantine on their physical activities tend to have higher prevalence of anxiety and depression symptoms and keeping active can play a very important protective factor of mental and physical health [14,15].

However, stress responses, self-care or health behaviour changes during COVID-19 lockdown can differ by age, gender, ethnicity, and socioeconomic position. This situation seems particularly stressful for younger adults (<35 years), women, people without work and with low incomes [16]. Moreover ethnic minorities showed to undertake less exercise and consume lower amounts of fruit and vegetables during lockdown. Regarding countries differences, it has been found that living in a high-income country during the pandemic is a risk factor for depression and anxiety and countries belonging to the Latin America and Caribbean cluster showed a lower prevalence of mental health symptoms compared to countries belonging to North America, Europe and Central Asia, and Sub-Saharan Africa clusters [17].

Nevertheless, to our knowledge, none have explored these differences in a well-balanced sample by age, gender, and country and neither on variables such as, people’s adherence to Public Health guidelines, stress perception, and the adoption of self-care activities. Therefore, in this study we hypothesized the existence of differences in responses to stress, health practices and self-care activities depending on country, age, and gender due to mandatory COVID-19 confinement. 

## 2. Materials and Methods 

### 2.1. Study Sample

The sample was obtained by an online survey shared on social media from 31 March to 14 April 2020. Participation was voluntary, anonymous, no compensation was offered, and the administration of the instruments (Perceived Stress Scale: PSS-10, COVID-19 Health Practices, and Self-Care Activities Screening Scale: SASS-14) took approximately 15 min. Approval was obtained from the Research Ethics Committee of the university responsible for the study (Project ID: 2020.058). The estimation of the sample size was based on the application of the central limit theorem, which states that when a sample exceeds 30 individuals, whatever the sample mean, it will approximately follow a normal distribution. Given that the statistical analyzes respond to the general linear model, the estimation of the sample would be calculated as follows: number of subjects (30) * number of age groups (5) * number of genders (2) * number of countries (4) = 1200. Out of 3452 respondents, a stratified sample was extracted by randomizing cases by the four countries, gender, and age groups. The final sample was comprised of 1082 participants.

### 2.2. Measures

#### 2.2.1. Perceived Stress 

The Spanish version of the PSS-10 evaluates perceived stress during the last month [18]. It is composed of 10 items ranging from 0 = Never to 4 = Very often. The instrument provides a total score; the higher the total score, the greater the level of perceived stress. The scale has good reliability (internal consistency, α = 0.81), concurrent validity, sensitivity, and Cronbach’s alpha for this sample was α = 0.85 [18]. Studies have been published that report in relation to PSS-10, optimal psychometric properties, both in the general population and people exposed to confinement [19,20] and specifically in health professionals who attend the emergency situation [21].

#### 2.2.2. COVID-19 Health Practices

A short ad hoc scale was used to examine three preventive health indicators during COVID-19 lockdown. A single item examined COVID-19 seriousness perception: “How serious do you consider this pandemic situation?” The response scale ranged from 1 = Not serious at all to 4 = Very serious. Two items ranging from 1 = Do not agree at all to 6 = Totally agree examined healthy daily routines and adherence to Public Health guidelines: “I do follow Public Health guidelines” and “I am keeping a healthy routine with balanced schedules and different places to work and rest”, respectively.

#### 2.2.3. Self-Care Activities

The SASS-14 is a brief self-report instrument to screen self-care activities in the general population [22]. This screening tool measures four dimensions: health consciousness, nutrition and physical activity, sleep quality and inter and intrapersonal coping strategies. The SASS-14 is composed of 14 items ranging from 1 = Never to 6 = Always. The higher the total score, the more self-care activities the person engages in. The SASS-14 has adequate psychometric properties with high internal consistency and convergent validity with psychological well-being and stress measures. For this sample, Cronbach’s alpha was α = 0.77.

### 2.3. Statistical Analysis

SPSS version 24 was used for data entry and analyses. Descriptive statistics analysis was used to summarize the socio-demographic data. Preliminary multiple regression analyses were conducted to determine whether socio-economic variables were associated with the main variables (i.e., stress, health practices, and self-care). Such analyses did not show significant influences of any socio-demographic variables on health practices indicators and self-care. On the other hand, educational level, income level, changes in the employment situation, being accompanied during lockdown, or having community resources showed a significant influence on stress (*R*² = 0.10; *p* < 0.001). Finally, only the income level variable had a significant influence in a preliminary univariate on stress. Therefore, analyses of the relationship between the main variables and factors were conducted using multivariate analysis (MANOVA), where health practices and self-care variables were included as dependent variables, and country, gender and age were included as independent variables. Stress, controlling for income level, was analyzed with a univariate analysis of covariance (ANCOVA). 

When significant interactions were found, the file was split by the responsible factors of the interaction and, subsequently, independent ANOVAs were conducted with each dependent variable. Bonferroni adjusted all multiple comparisons and partial eta square (ηp^2^) was used for test effect size.

## 3. Results

### 3.1. Socio-Demographic Characteristics

Data was obtained from 1082 participants recruited in four countries: Spain (n = 271, with a ratio of the sample participation relation to the population of legal age of 0.0007%, source: INE), Ecuador (n = 282, 0.0024%, source: INEC), Chile (n = 261, 0.0024%, source: INE), and Colombia (n = 268, 0.00081%, source: DANE). Participants’ age ranged from 18 to 95 with a mean age of 43.9 (*SD* = 15.2); 50.9% (551) of the sample was female and 49.1% (531) male. Socio-demographic characteristics were similar across countries with very few differences between genders, most participants having a high educational level and medium and high-income level, as shown in Table 1.

### 3.2. Perceived Stress

After correcting for income level, the univariate ANCOVA analysis showed no significant interactions among country, gender, and age group (*F*(12) = 0.75; *p* = 0.70). However, a significant interaction was found between gender and age group (*F*(4) = 2.76; *p* = 0.03; *ηp*^2^ = 0.01) due to the fact that females from the youngest group showed significantly higher levels of stress compared to females from older groups (*p* = 0.02; *p* = 0.005; *p* = 0.001). Females from the 29–39 age group showed significantly higher stress levels than those over 60 (*p* = 0.01). These differences are also significant between females and males from the youngest age groups (18–28 years old) as young females reported higher levels of stress compared to young males *(p* = 0.002). Main effects analysis shows that differences in terms of perceived stress across countries are significant (*F*(3) = 6.92; *p* < 0.001; *ηp*^2^ = 0.02), those from Chile and Spain having higher stress levels compared to Colombian people (*p* = 0.001; *p* < 0.001), as shown in Figure 1A.

### 3.3. COVID-19 Health Practices

#### 3.3.1. COVID-19 Seriousness Perception

The interaction found with regards to COVID-19 seriousness perception among country, gender, and age groups (*F*(12) = 0.83; *p* = 0.62) was not significant. However, a significant interaction between gender and age groups was found in relation to this indicator (*F*(4) = 5.04; *p* < 0.001; *ηp*^2^ = 0.20). Male participants from the youngest group (18–28 years old) reported greater seriousness perception than males from the 50–59 (*p* = 0.003) and >60 age groups (*p* = 0.009). Similarly, males from the middle age group (40–49) showed greater seriousness perception than those aged from 50 to 59 and >60 (*p* < 0.001; *p* < 0.001). Females from the youngest group reported less seriousness perception compared to males of the same age (*p* = 0.01). However, females from the older group (50–59) reported higher scores than males from this age group (*p* = 0.002), as shown in Figure 1B.

#### 3.3.2. Adherence to Public Health Guidelines

A significant interaction was found among country, gender, and age groups in the case of adherence to health guidelines (*F*(12) = 2.74; *p* = 0.001; *ηp*^2^ = 0.30). Inter-gender by age and country analysis showed significant differences for Colombian females in the 40–49 age group, who showed less adherence than their male peers (*p* = 0.03). In contrast, Colombian females in the 50–59 age group reported greater adherence than males of the same age (*p* = 0.01). Meanwhile, Ecuadorian females from the youngest and middle age groups (18–59) showed less adherence than males of the same age (*p* = 0.01; *p* = 0.01; *p* = 0.04). Spanish females aged between 29 and 39 also reported lower scores than males (*p* = 0.02). Similar to Colombian participants, Ecuadorian females in the 50–59 age group reported higher scores than their male peers (*p* = 0.04). Chilean females over the age of 60 showed lower scores than males (*p* = 0.02) while Spanish females over the age of 60 reported greater adherence (*p* = 0.02).

Inter-age by gender and country analysis showed that Ecuadorian females aged between 18 and 28 obtained lower adherence scores than those from the 40–49 and 50–59 age groups in the same country (*p* = 0.04; *p* = 0.003). Similarly, 40–49 year old Spanish females reported higher scores than those from the 50–59 age group (*p* = 0.02). This last group showed lower scores compared to females over the age of 60 (*p* = 0.03). Regarding males, those aged over 60 in Spain reported lower adherence scores than those from the 29–39 and 40–49 age groups (*p* = 0.01; *p* = 0.02). Inter-country by gender and age analysis showed that 50–59 year old females from Colombia reported higher levels of adherence than Spanish females (*p* = 0.01). Similarly, Ecuadorian females from the same age group showed higher scores than Spaniards (*p* = 0.004). Colombian males in the 50–59 age group reported lower scores than Chilean males of the same age (*p* = 0.04). For males over 60, results showed that Spanish males reported lower scores than Chilean males of the same age (*p* = 0.04), as shown in Figure 1C. 

#### 3.3.3. Healthy Daily Routines

Regarding the practice of healthy daily routines, a three-factor interaction was found among country, gender, and age group (*F*(12) = 3.71; *p* < 0.001; *ηp*^2^ = 0.40). In the case of inter-gender by age and country analysis, results showed that 29–39 and 50–59 year old females from Colombia reported higher levels of healthy daily routines than males (*p* < 0.001; *p* = 0.02). In contrast, 50–59 and >60 year old Chilean males showed higher scores than females (*p* = 0.01; *p* = 0.006). Spanish females over the age of 60 reported higher levels than males (*p* = 0.008).

Inter-age by gender and country analysis showed that young females from Colombia reported lower levels than those from the 50–59 age group (*p* = 0.05). These differences were also detected in Spain but in the middle age group, where 40–49 year old females reported healthier routines than those from the 50–59 group (*p* = 0.03). This last group showed a significantly higher level than females over 60 (*p* = 0.004). However, young Spanish males (29–39) reported higher scores than those aged between 50 and 59 (*p* = 0.05).

Inter-country by age and gender analysis showed that 29–39 year old Colombian females reported higher levels than Ecuadorian females of the same age (*p* = 0.03). Colombian females aged between 50 and 59 showed higher levels than Spanish and Chilean females (*p* = 0.02; *p* = 0.02). Spanish females over 60 reported higher levels than Chileans (*p* = 0.02). Regarding males, Chilean and Spanish males from the 29–39 age group reported higher scores than Colombians (*p* = 0.04; *p* < 0.001), as shown in Figure 2A.

### 3.4. Self-Care Activities

With regards to the total score of self-care activities, the univariate analysis showed no interactions among country, gender and age range for this variable (*F*(12) = 1.19; *p* = 0.29; *ηp*^2^ = 0.01). However, a main effect was detected for gender (*F*(1) = 7.00; *p* = 0.008; *ηp*^2^ = 0.01), which can be explained by the fact that females in general scored higher in self-care activities than males (*p* = 0.008), as shown in Figure 2B.

To further examine differences between groups, a second multivariate analysis was conducted on the health consciousness dimension as a key element of self-care. 

#### Health Consciousness

An interaction was detected for this indicator between gender and country (*F*(3) = 3.08; *p* = 0.02; *ηp*^2^ = 0.01). Differences were found for Colombian and Ecuadorian females, who showed a lower level of health consciousness compared to males (*p* = 0.007; *p* < 0.001). An interaction was also found between gender and age group (*F*(4) = 2.92; *p* = 0.02; *ηp*^2^ = 0.01). Females from the youngest group reported a significantly lower level of health consciousness than the older group (50–59 and above 60) (*p* = 0.05; *p* < 0.001). Meanwhile, males from the youngest groups reported lower scores than the older groups (29–59) (*p* = 0.05; *p* = 0.002; *p* = 0.03). Inter-gender analysis showed significant differences, as 18 to 59 year old females reported lower levels than their male peers (*p* = 0.03; *p* = 0.02; *p* = 0.003; *p* = 0.05). In contrast, no significant differences were found between females and males over 60, as shown in Table 2.

## 4. Discussion

As it was hypothesized, we found different responses to stress, health practices, and self-care according to country, age, and gender. Our results suggest that regardless of the country and controlling for income level, females from the youngest age group suffered greater levels of stress, showed a lower level of adherence to health guidelines and reported lower levels of health consciousness, in comparison to their male peers and older groups. However, regarding self-care, it seems that females are generally more involved in self-care activities and adopt healthier daily routines than males. Likewise, despite young males perceiving this situation as more severe than their female peers, this result is inverted for females over 50 who reported higher scores than males. The same happened in the case of adherence to health guidelines, as females from the age of 50 showed higher levels than their male peers.

These results were mostly consistent among Colombia, Ecuador, and Spain. However, Chile showed some different tendencies as 29–39 year old males showed healthier daily routines than females and better adherence to health guidelines than people from 50 to 60 years old. Similarly, females from Colombia and Ecuador aged over 50 showed greater adherence to health guidelines than their Spanish peers. With regards to health consciousness, significantly lower levels in young females were especially noticeable in Colombia and Ecuador. 

Our findings are in line with several studies conducted in Spain, Austria, and UK, which have found that this situation seems to have a higher impact on women and young people, particularly stressful for those <35 years, people without work, and low income [4,16,17,18,19,20,21,22,23].

Regarding country differences, they could be explained by the fact that the average number of days that people were confined in Spain and Ecuador was higher than in Colombia and Chile at the time of the survey (March to April), which is in line with other studies where Latin America and Caribbean clusters showed a lower prevalence of mental health symptoms at that time [17]. 

It may be because of the impact of the coronavirus disease on South America was not as severe as it was in Spain at that point according to epidemiological data from Johns Hopkins University website [1]. In addition, Chile reported a lower percentage of males in charge of children and older people, which may also be a factor in the lower levels of stress suffered by this group, thus enabling them to maintain healthier daily routines and report better adherence to health guidelines [2]. 

Regarding stress, our findings are in line with other studies that have found that emotional well-being worsens during COVID-19 lockdown [6]. In particular, our results on gender and age differences are similar to those that have demonstrated that stress has increased during lockdown and that females and the youngest groups are the most affected [4,24]. These findings could be related to the fact that young adults perceived this situation as more severe than older people did. However, young adults reported a lower level of adherence to health guidelines than the older groups. Therefore, this perception of seriousness in young people might be more greatly associated with the impact of the pandemic on their personal situation (i.e., working or studying situation, social life, or changes to their lifestyle) rather than to their perception of the health risks. It may be critical to explore health and risk communications since, as a recent study has suggested, people perceiving greater risks are more likely to implement protective behaviors—especially later (versus earlier) in time [25]. However, these risks may be perceived differently across age ranges and may be different between women or men depending on their personal situation. 

Concerning gender, the fact that women are reporting higher scores on perceived stress was an expected outcome consistent with previous studies during lockdown [26]. However, this differential impact of gender on mental health outcomes goes beyond the pandemic situation, and could be argued to stem from women’s roles, mainly the burden of both work and caring roles, as well as to the number of social roles women fulfil [27]. These gender differences may also contribute to a greater vulnerability for women, not only by contracting the virus, given that the majority of workers in frontline sectors are women, but also because of an overload of their caring role and therefore, an experience of greater stress levels and mental health problems [28]_._

In relation to self-care, it has been considered either in clinical samples or general population, as an important factor affecting one’s self-care abilities, perceived control, and knowledge of self-care behaviors [29,30]. In this study, women from the middle age groups onwards seemed to engage in more self-care activities and adopt healthier daily routines than males. This result is in line with previous studies which have supported the mediation role of stress in the relationship between gender and health-promoting behaviors [27]. Whose results indicated that while women report a wider variety of health promoting-behaviour than men, they might refrain from those behaviors because of their levels of general stress, which could be the case of the current pandemic. This fact may also explain why in our study the youngest females whose stress levels were high, reported also less self-care behaviors than older groups. This result is similar to other studies during the confinement, indicating a direct relationship between physical activity, gender, and mental health outcomes [26,31]. Similar to stress and gender discussion, these results might be likely embedded in gendered role behavior as women adopt more caregiving behaviors and use a wider range of self-care activities than men [32]. Therefore, health recommendations may need to focus on gender-informed approaches when health-promoting behaviors or self-care.

Nevertheless, these differences went in the opposite direction for the health consciousness dimension, as men seemed to be more health-conscious than women. Because of the nature of this dimension, this result may be linked more to their seriousness perception and adherence to health guidelines scores, rather than the adoption of self-care activities. Thus, gender differences can vary depending on the dimension of self-care explored. Future research should further examine gender differences with regards to the assessed variables to ascertain their nature.

Concerning age, the fact that older people showed better adherence to health guidelines, health consciousness, and other self-care activities (in comparison with the younger groups) may be explained by two main reasons. On the one hand, the unavoidable differences in lifestyles across age. The daily lives of older adults are less diverse in terms of their social interaction partners. However, older adults report greater diversity in activities compared to younger adults as they must work or study [33]. These differences could be even more acute within a lockdown experience since confinement has significantly affected the social and studying/working habits and daily routines of young people. 

On the other hand, health consciousness plays a key role in the adoption of health behaviors in young adults [34]. According to a previous study, older adults (60–92) are more likely to engage in more health-responsible behaviors and score higher in health-promoting behaviors than middle-aged adults (40–59) and younger ones (18–39) [35]. These differences could also be explained by the fact that general health guidelines are not being age-targeted in most countries. Most informative and mass media are not adapting their messages to the worries, needs, and resources of young adults and adolescents. Thus, they may not be as health conscious as older people are. Moreover, as a recent qualitative study showed, self-care strategies within the young population are mostly based on social media resources [36], whereas these resources are not being used enough to disseminate adherence to health practices and self-care behavior in these age groups. 

Therefore, it may be critical to increase health promotion and education through digital resources and social media [37]. Moreover, since young people are very influenced by social norms, it would be essential to adapt health communication to their social identities, in order to promote social norms and behaviors based on accurate information [38]. However, it should be done without using fear appeals as a health communication strategy [39]. An alternative way to do this might be to focus public messages on the positive health behaviors that people are adopting, rather than focusing on the undesirable ones [40]. Nevertheless, some of these behaviors are also greatly influenced by trust in the government [41]. Thus, differences in national attachment across age groups could influence these results. 

It is also noteworthy that this lack of adherence is often condemned as irresponsible and selfish, however changing people’s behaviour is simply not as easy as just informing them of the risks. As Van den Broucke [42] highlights, it is well-known in health promotion models it is profoundly linked to the fact that people may not consider themselves at risk (e.g., if they have not been in contact with others who have been contaminated), may underestimate the seriousness of the condition (e.g., when they are told that most fatalities are older people or people with pre-existing morbidity) or may not see themselves as capable to perform the preventive behaviors [42].

### Limitations

Firstly, this study included people with a similar high socio-economic situation in the four countries that composed our sample. Therefore, these findings may not be representative for more disadvantaged or vulnerable social groups. Secondly, self-report instruments were used, where social desirability may have influenced these results and also these instruments were only in the Spanish language so it may not capture the full language diversity of each participant country. However, questions related to sociodemographic data, self-care, and health activities sections, were adapted by eight judges (two for each country) to standardize into Spanish from each of the participating countries. Thirdly, the study has a cross-sectional design, and thus it is not possible to conclude causal relations between the assessed variables. Lastly, the sample size of the factor age groups could have been too small to detect bigger differences. Another limitation was that due to the characteristics of the sample, there is a different percentage of participation of the population by country, this percentage being higher in Chile and Ecuador. 

## 5. Conclusions

Due to the unquestionable importance of promoting healthy behaviors and self-care in general, especially during a lockdown, socio-economic, age, and gender differences should be considered when addressing health recommendations. Developing strategies to get the most vulnerable groups involved in health behaviors, as well as reinforcing those that serve as a role model, could increase the success of people´s adherence to health guidelines and self-care. This would lead to improving the health, wellness, and well-being of individuals, thus reducing the high costs of medical services. Therefore, it is mandatory to optimize public policies to make them more health promoting, taking into account the social determinants of health and by altering social norms so that the health of all members and groups of society is a priority. In order to do so, some implications for research, policy and practice are described below.

### 5.1. A More Integrative Health Promotion Approach

On one hand, in the same vein that research has noticeable increased on the impact of the COVID-19 pandemic on mental health outcomes, more systematic research is needed to understand the relationship between health behaviors and mental health outcomes to better understand the short- and long-term consequences of this mental health crisis and explore more comprehensives approaches to address it [43,44]. On the other hand, health policies, measures and media are needed to promote greater health behaviors with a special emphasis on health consciousness and self-care. Enabling people to increase control over their health and its determinants is at the core of health promotion, which paradoxically is more important in this time of crisis than ever before [42]. This health promotion approach can contribute at different levels [45] the downstream level focusing on individual behaviour change and disease management, at the midstream level through interventions affecting organizations and communities and at the upstream level through informing policies affecting the population. As Van den Broucke [42] as pointed out, the expertise with regard to health behaviour change is one of the core competencies of health educators and promoters, and their advice may help governments to achieve the required behaviour change.

### 5.2. A Gender-Based Approach

Since the impact of COVID-19 pandemic and physical distancing measures on health behaviors drive important health inequalities, especially in those disadvantaged groups, further studies should monitor the differential impacts of the current pandemic across age, gender, socioeconomic disadvantage (in early and adult life) and culture/ethnicity and their possible implications to population health and the widening of health inequalities [16]. However, in order to reduce these health inequities and address immediate and long-term consequences, it is urgent to establish strategies for public health emergencies that take a gender-based approach into account [46,47].

### 5.3. Intersectionality and Community Action

While preventing the further spread of COVID-19 relies heavily on informing and encouraging the population to adopt protective behaviors, these efforts may be more successful if the advice from experts is combined with local community knowledge [42] and intersectoral strategies [47]. According to The Ottawa Charter, it is crucial in health promotion strategies to emphasize the importance of community action, in the sense of needs assessments, setting priorities, joint planning, capacity building, strengthening local partnerships, intersectoral working, and enhancing public participation and social support [48]. All of these activities are aimed to create empowered communities, where individuals and organizations apply their skills and resources in collective efforts to address health priorities and meet their respective health needs. Therefore, community engagement can make a substantial difference in health outcomes and strengthen the capacity to deal with the negative consequences of the pandemic at individual, organizational, and community level.

## Figures and Tables

**Figure 1 ijerph-18-02253-f001:**
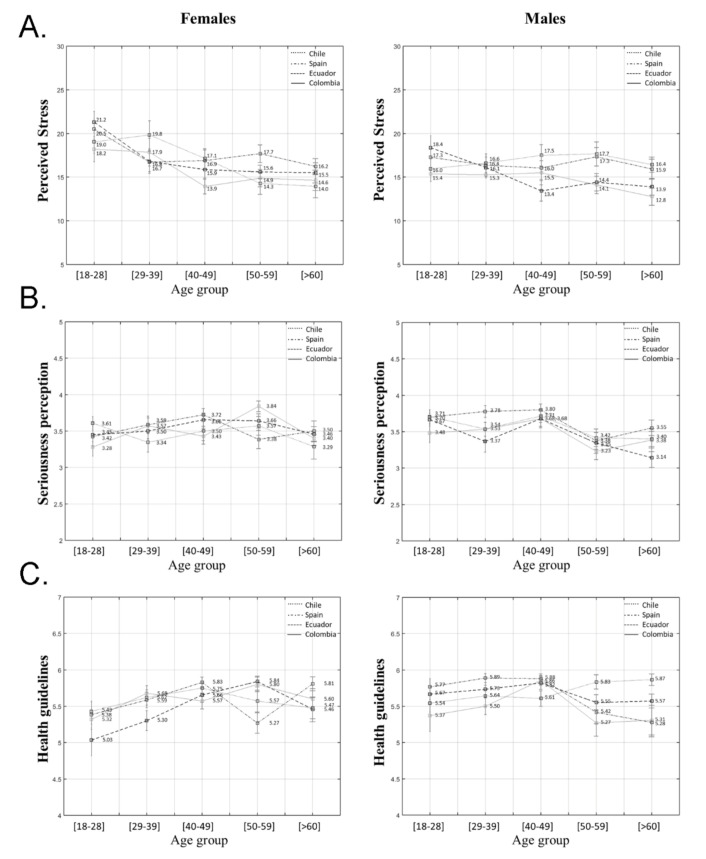
Values for (**A**) perceived stress, (**B**) seriousness perception, and (**C**) health guidelines adherence differentiated by gender, age group, and country.

**Figure 2 ijerph-18-02253-f002:**
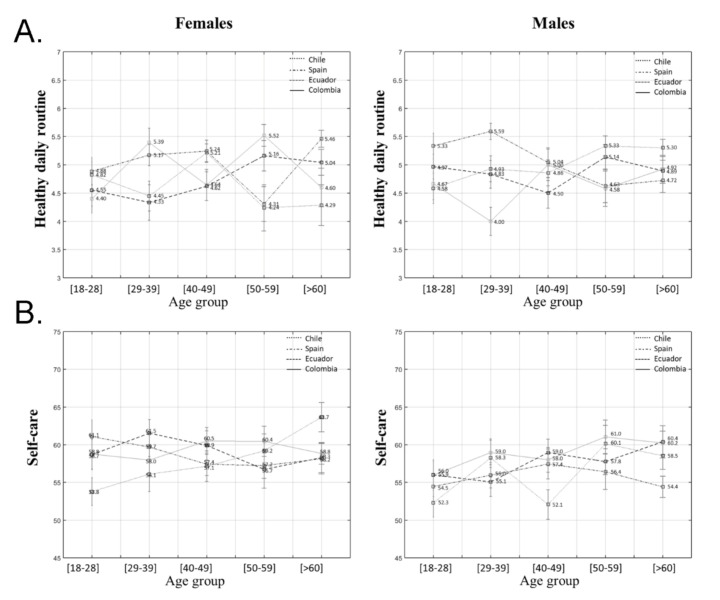
Values for (**A**) healthy daily routine, and (**B**) self-care differentiated by gender, age group, and country.

**Table 1 ijerph-18-02253-t001:** Sample age-range distribution and socio-demographics characteristics by country and gender.

	Spain					Chile					Colombia					Ecuador					
	18–28	29–39	40–49	50–59	<60	18–28	29–39	40–49	50–59	<60	18–28	29–39	40–49	50–59	<60	18–28	29–39	40–49	50–59	<60	
	57(21)	55(20)	54(20)	50(18)	55(20)	56(21.5)	53(20.3)	56(21.5)	45(17)	51(19.5)	56(21)	54(20)	56(21)	51(19)	51(19)	64(23)	55(19.5)	57(20)	54(19.1)	52(18)	
	Spain					Chile					Colombia					Ecuador					
	Female		Male		Total	Female		Male		Total	Female		Male		Total	Female		Male		Total	Full Sample
N (%)	136(50.2)		135(49.8)		271	127(48.7)		134(51.4)		261	131(48.9)		147(51.1)		268	137(48.6)		145(51.4)		282	1082
**Income n (%)**																					
No salary	22(16.2)		23(17)		45(16.6)	19(15.0)		8(6.0)		27(10.3)	21(16)		22(16.1)		43(16.0)	23(16.8)		20(13.8)		43(15.2)	158(14.6)
One mw	4(2.9)		5(3.7)		9(3.3)	13(10.2)		9(6.7)		22(8.4)	21(16)		16(11.7)		37(13.8)	12(8.8)		7(4.8)		19(6.7)	87(8)
Two mw	20(14.7)		10(7.4)		30(11)	21(16.5)		15(11.2)		36(13.8)	19(14.5)		18(13.1)		37(13.8)	26(19.0)		16(11.0)		42(15)	145(13.4)
Three mw	28(20.7)		22(16.3)		50(18.5)	14(11.0)		21(15.7)		35(13.4)	34(26.0)		34(24.8)		68(25.3)	26(19.0)		10(6.9)		36(12.7)	189(17.5)
Four mw	30(22.1)		29(21.6)		59(21.7)	15(11.8)		19(14.2)		34(13)	14(10.7)		16(11.7)		30(12.0)	26(19.0)		26(17.9)		52(18.4)	175(16.2)
Five mw	32(23.5)		46(34.1)		78(28.8)	45(35.4)		62(46.3)		107(41)	22(16.8)		31(22.6)		53(19.7)	24(17.5)		66(45.6)		90(32)	328(30.3)
**Educational Level n (%)**																					
Elementary	1(0.7)		3(2.2)		4(1.48)	2(1.6)		2(1.5)		4(1.53)	0(0.0)		1(0.7)		1(0.4)	2(1.5)		0(0)		2(0.7)	11(1)
High School	20(14.7)		22(16.3)		42(15.5)	12(9.4)		16(11.9)		28(7.0)	14(10.7)		21(15.3)		35(13.5)	20(14.6)		28(19.3)		48(17.0)	153(14.1)
Technical	18(13.2)		19(14.1)		37(13.7)	18(14.2)		15(11.2)		33(12.7)	16(12.2)		24(17.5)		40(15.3)	7(5.1)		5(3.4)		12(4.3)	122(11.3)
University	97(71.3)		91(67.4)		188(69.3)	95(74.8)		101(75.4)		196(75.1)	101(77.1)		91(66.4)		192(74.5)	108(78.8)		112(77.2)		220(78.0)	796(73.6)
**COVID-19 variables n (%)**																					
Frontline workers (yes)	34(25)		46(34.1)		80(29.5)	36(28.3)		44(32.8)		80(30.7)	70(53.4)		61(44.5)		131(48.9)	35(25.5)		41(28.3)		76(27)	367(33.9)
Health risk factors (yes)	39(28.7)		50(37)		89(32.8)	33(26)		51(38.1)		84(32.2)	44(33.6)		33(24.1)		77(28.7)	31(22.6)		56(38.6)		87(30.9)	337(31.1)
Employment changes (yes)	21(15.4)		25(18.5)		46(17)	32(25.2)		19(14.2)		51(19.5)	41(31.3)		46(33.6)		87(32.4)	41(30)		46(31.7)		87(31)	271(25)
Accompanied during lockdown (yes)	114(83.8)		117(86.7)		231(85.2)	117(92.1)		120(89)		237(91)	130(95)		136(94)		266(99.2)	121(92.4)		127(92.7)		248(88)	982(90.8)
Community resources (yes)	120(88.2)		125(92.6)		245(90.4)	102(80.3)		113(84.3)		233(89.2)	109(83.2)		116(85)		225(84)	118(86.1)		133(91.7)		251(89)	936(85.5)
Children in charge (yes)	32(23.5)		31(23)		63(23.2)	48(37.8)		38(28.4)		86(33)	37(28.2)		56(41)		93(34.7)	47(34.3)		48(33.1)		95(33.6)	337(31.1)
Older people in charge (yes)	15(11)		14(10.4)		29(10.7)	37(30)		20(15)		57(22)	53(40.5)		53(38.7)		106(39.5)	47(34.3)		48(33.1)		95(33.6)	287(26.5)
**Confinement days** **M (SD)**	21(4.6)					17.5(6.5)					17(4.0)					25(0.6)					

Note: mw = minimum wage.

**Table 2 ijerph-18-02253-t002:** ANOVA’s for pair means comparison of main variables interaction.

Variable (Likert Scale)	ANOVA	Mean Comparison	Dif _Mean_	Error	*p* Value	95% CI	
Perceived stress (1–6)	Inter-gender by age, *F*(4) = 2.88, *p* = 0.02	Female 18–28 vs. male	20.96	0.92	0.002	10.34	40.78
	*F*(4) = 6.71, *p* < 0.001	Female 18–28 vs. 40–49	30.12	0.89	0.005	0.62	50.62
		Female 18–28 vs. 50–59	30.47	0.91	0.002	0.91	60.04
		Female 18–28 vs. <60	30.96	0.92	0.000	10.37	60.56
		Female 29–39 vs. <60	20.62	0.86	0.024	0.19	50.04
	*F*(4) = 6.83, *p* < 0.001	Colombia vs. Chile	−20.03	0.55	0.001	−30.47	−0.59
		Colombia vs. Spain	−20.17	0.54	0.000	−30.59	−0.76
Seriousness perception of COVID-19 pandemic (1–4)	Age x gender *F*(4) = 8.14, *p* < 0.001	Males 18–28 vs. 50–59	0.29	0.81	0.003	0.07	0.53
		Males 18–28 vs. >60	0.28	0.004	0.009	0.04	0.49
		Males 40–49 vs. 50–59	0.37	0.08	<0.001	0.14	0.61
		Male 40–49 vs. >60	0.34	0.08	<0.001	0.12	0.57
	*F*(1) = 6.44, *p* = 0.01	Female 18–28 vs. Male	−0.20	0.07	0.012	−0.34	−0.04
	*F*(1) = 10.15, *p* = 0.002	Female 50–59 vs. Male	0.27	0.08	0.002	0.10	0.43
Public health guidelines (1–6)	Inter-gender by age and country *F*(4) = 6.48, *p* = 0.01	40–49 Colombian females vs. males	−0.29	0.13	0.029	−0.54	−0.03
		50–59 Colombian females vs. males	0.53	0.21	0.014	0.11	0.95
	*F*(4) = 7.05, *p* = 0.01	18–28 Ecuadorian females vs. males	−0.63	0.24	0.010	−10.11	−0.16
		29–39 Ecuadorian females vs. males	−0.43	0.17	0.012	−0.77	−0.10
		50–59 Ecuadorian females vs. males	0.29	0.13	0.036	0.02	0.56
	*F*(4) = 6.27, *p* = 0.01	>60 Chilean females vs. males	−0.39	0.16	0.016	−0.70	−0.08
	*F*(4) = 5.90, *p* = 0.01	29–39 Spanish females vs. males	−0.30	0.12	0.018	−0.55	−0.05
	*F*(4) = 5.43, *p* = 0.24	>60 Spanish females vs. males	0.53	0.23	0.024	0.07	0.09
	Inter-age by gender and country *F*(4) = 4.24, *p* = 0.003	Ecuadorian females de 18–28 vs. 40–49	−0.62	0.21	0.035	−10.22	−0.03
		Ecuadorian females de 18–28 vs. 50–59	−0.81	0.22	0.003	−10.42	−0.19
	*F*(4) = 3.98, *p* = 0.004	Spanish females 40–49 females 50–59	−0.56	0.17	0.017	−10.06	−0.06
		Spanish females 40–49 females 50–59	−0.54	0.18	0.031	−10.05	−0.03
	*F*(4) = 4.50, *p* = 0.002	Spanish males 29–39 vs. males >60	0.61	0.19	0.012	0.08	10.14
		Spanish males 40–49 vs. males >60	0.60	0.19	0.017	0.06	10.14
	Inter-country by gender and age *F*(4) = 5.17, *p* = 0.002	50–59 Ecuadorian female vs. Spanish	0.57	0.16	0.004	0.13	10.01
		50–59 Chilean males vs. Colombian	0.56	0.20	0.035	0.03	10.10
		>60 Chilean males vs. Spanish	0.59	0.21	0.036	0.02	10.16
Healthy daily routines (1–6)	Inter-gender by age and country *F*(1) = 15.32, *p* < 0.001	29–39 Colombian females vs. males	10.39	0.36	<0.001	0.68	20.11
	*F*(1) = 6.42, *p* = 0.02	50–59 Colombian females vs. males	0.94	0.37	0.015	0.20	10.69
	*F*(1) = 6.63, *p* = 0.01	50–59 Chilean females vs. males	−10.10	0.43	0.014	−10.95	−0.24
	*F*(1) = 8.35, *p* = 0.006	>60 Chilean females vs. males	−10.01	0.35	0.006	−10.72	−0.31
	*F*(1) = 7.56, *p* = 0.008	>60 Spanish females vs. males	0.74	0.27	0.008	0.20	10.28
	Inter-age by gender a country, *F*(4) = 6.42, *p* = 0.02	Colombian females 18–28 vs. 50–59	−10.12	0.39	0.046	−20.23	−0.01
	*F*(4) = 3.92, *p* = 0.005	Spanish females 40–49 vs. 50–59	0.93	0.31	0.032	0.05	10.82
		Spanish females 50–59 vs. >60	−10.15	0.32	0.004	−20.07	−0.24
	*F*(4) = 3.09, *p* = 0.02	Spanish males 29–39 vs. 50–59	0.97	0.34	0.046	0.01	10.93
	Inter-country by age and gender *F*(1) = 4.08, *p* = 0.009	29–39 Colombian females vs. Ecuadorian	10.06	0.37	0.028	0.07	20.05
	*F*(4) = 4.57, *p* = 0.005	50–59 Colombian females vs. Spanish	10.21	0.41	0.022	0.11	20.31
		50–59 Colombian females vs. Chilean	10.28	0.43	0.022	0.12	20.44
	*F*(3) = 3.44, *p* = 0.02	Spanish females >60 vs. Chilean	10.18	0.40	0.023	0.11	20.24
	*F*(3) = 8.44, *p* < 0.001	Colombian males 29–39 vs. Chilean	−0.93	0.32	0.024	−10.78	−0.08
		Colombian males 29–39 vs. Spanish	−10.59	0.32	0.000	−20.45	−0.74
Health consciousness (1–6)	Inter-gender by country, *F*(1) = 7.36, *p* = 0.007	Colombian females vs. Colombian	−10.60	0.59	0.007	−20.76	−0.44
	*F*(1) = 13.80, *p* < 0.001	Ecuadorian females vs. males	−20.11	0.57	0.000	−30.23	−0.99
	Intra-gender by age *F*(4) = 6.57, *p* < 0.001	Female 18–28 vs. female 50–59	−10.91	0.67	0.047	−30.80	−0.02
		Female 18–28 vs. female >60	−30.44	0.67	<0.001	−50.33	−10.54
	*F*(4) = 4.09, *p* < 0.001	Males 18–28 vs. males 29–39	−10.75	0.62	0.050	−30.49	0.00
		Males 18–28 vs. males 40–49	−20.32	0.63	0.002	−40.09	−0.55
		Males 18–28 vs. males 50–59	−10.89	0.64	0.031	−30.69	−0.10
	Inter-gender by age, *F*(1) = 4.69, *p* = 0.03	Females 18–28 vs. males 18–28	−10.35	0.62	0.031	−20.58	−0.12
	*F*(1) = 5.58, *p* = 0.02	Females 29–39 vs. males 29–39	−10.42	0.60	0.019	−20.61	−0.24
	*F*(1) = 9.18, *p* = 0.003	Females 40–49 vs. males 40–49	−10.87	0.62	0.003	−30.09	−0.66
	*F*(1) = 3.96, *p* = 0.05	Females 50–59 vs. males 50–59	−10.34	0.67	0.048	−20.66	−0.01
Nutrition and physical activity (1–6)	Age main effect *F*(4) = 4.66, *p* = 0.001	18–28 vs. 29–39	10.21	0.41	0.029	−20.35	−0.07
		18–28 vs. >60	10.39	0.42	0.009	0.21	20.56
		40–49 vs. >60	10.19	0.42	0.041	0.03	20.37
	Country main effect, *F*(3) = 4.16, *p* = 0.006	Ecuador vs. Spain	−10.24	0.37	0.005	−20.21	−0.27
Sleep (1–6)	Inter-age by country *F*(4) = 3.55, *p* = 0.008	Chilean 29–39 vs. 40–49	−10.35	0.45	0.031	−20.63	−0.70
		Chilean 29–39 vs. >60	−10.41	0.46	0.025	−20.72	−0.10
	*F*(4) = 2.99, *p* = 0.02	Ecuadorian 18–28 vs. 29–39	10.33	0.43	0.021	0.12	20.54
	Inter-country by age *F*(3) = 4.66, *p* = 0.003	Ecuadorian 29–39 vs. Spain 29–39	−10.53	0.50	0.005	−30.01	−0.36
		Chilean 29–39 vs. Spain 29–39	−10.53	0.50	0.016	−20.87	−0.18
Intra and interpersonal coping skills (1–6)	Intra-gender by age, *F*(4) = 3.23, *p* = 0.01	Female 40–49 vs. >60	−10.57	0.47	0.008	−20.88	−0.25
	*F*(4) = 2.41, *p* = 0.05	Male 18–28 vs. >60	10.40	0.48	0.034	0.06	20.74
	Inter-gender by age, *F*(1) = 10.94, *p* = 0.001	>60 Female vs. male	10.62	0.49	0.001	−0.65	20.58
	Country main effect, *F*(3) = 2.98, *p* = 0.03	Ecuador vs. Spain	−10.24	0.37	0.005	−20.21	−0.27

## Data Availability

The data presented in this study are available on request from the corresponding author. The data are not publicly available due to data privacy was agreed with study participants through informed consent.

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
