# Peer review of "Different Responses to Stress, Health Practices, and Self-Care during COVID-19 Lockdown: A Stratified Analysis"

_ijerph, 2021, doi:10.3390/ijerph18052253_

Round 1

Reviewer 1 Report

This study aimed to analyze the differential impact of the first COVID-19 lockdown on stress, health practices, and self-care activities among four countries (Spain, Chile, Colombia, and Ecuador). Important findings are presented, however, the study must be improved. Please, find in the attached file some suggestions. 

Author Response

Dear Reviewer

We would like to thank the editor and the referees for their expeditious and professional review and helpful comments. We answered all their questions and modified the manuscript according to their suggestions. The reviewers' advice have led us to improve the revised version of the manuscript. We have highlighted in yellow the paragraphs that imply changes in the new version of the manuscript.

Reviewer 1

Comments (C) for Authors

This study aimed to analyze the differential impact of the first COVID-19 lockdown on stress, health practices, and self-care activities among four countries (Spain, Chile, Colombia, and Ecuador). Important findings are presented, however, the study must be improved. Please, find in the attached file some suggestions.

R1. Page 1, line 18: Change 1082 to One thousand and eigthy-two.

R1C1: We want to thank the reviewer for this comment.

R1. Page 2, line 66: I suggest to add a hypothesis at the end of the Introduction section.

R1C2. We have introduced the hypothesis in the end of the Introduction section (R1C2). “Therefore, in this study we hypothesized the existence of differences in responses to stress, health practices and self-care activities depending on country, age and gender due to mandatory COVID-19 confinement”.

R1. Page 2, line 68: I suggest to add information about the calculation of the sample size.

R1C3. The estimation of the sample size was based on the application of the central limit theorem, which states that when a sample exceeds 30 individuals, whatever the sample mean, it will approximately follow a normal distribution. Given that the statistical analyzes respond to the general linear model, the estimation of the sample would be calculated as follows: number of subjects (30) * number of age groups (5) * number of genders (2) * number of countries (4) = 1200. The final number of participants in the present study was 1082 which is close to the estimated sample size. According to the reviewer suggestion, this information has now been added to the final version of the document.

R1. Page 2, line 71: To add more information about the "instruments" used in the study.

R1C4. We have introduced in that section the tests implemented and the titles of the measurement section have been modified.

“The sample was obtained [...], and the administration of the instruments (Perceived Stress Scale: PSS-10, COVID-19 Health Practices, and Self-Care Activities Screening Scale: SASS-14) took approximately 15 minutes.

R1. Page 3, line 120: I suggest to add some information about the population of each country, and if possible, to determine a ratio between the number of participants in the study and the total population of each country. Moreover, it would be interesting to consider these findings in the Discussion section. 

R1C5. The relationship between the number of participants and the population over 18 years of age in each country has been calculated according to data disposable in the official institutions of each country: Colombia has 33,241,321 inhabitants of legal age, so the sample would represent a 0.00081%, source: Administrative Department National Statistics Office (DANE); Spain has 38,994,323 inhabitants of legal age, so the sample would represent a 0.0007%, source: National Institute of Statistics (INE-Spain); Chile has 10,931,721 inhabitants of legal age, so the sample would represent a 0.0024%, source: National Institute of Statistics (INE-Chile), and Ecuador has 11,511,613 inhabitants of legal age, so the sample would represent a 0.0024%, source: National Institute of Statistics (INEC-Ecuador). We have included these proportions in the sociodemographic characteristics in the body of the manuscript.

“Data was obtained from 1,082 participants recruited in four countries: Spain (n = 271, with a ratio of the sample participation relation to the population of legal age of 0,0007%, source: INE), Ecuador (n = 282, 0,0024%, source: INEC), Chile (n = 261, 0,0024%, source: INE) and Colombia (n = 268, 0,00081%, source: DANE).

Moreover, as the reviewer suggests, we have included this limitation in the discussion section.

“Another limitation was that due to the characteristics of the sample, there is a different percentage of participation of the population by country, this percentage being higher in Chile and Ecuador”.

R1. Page 3, line 125: To connect Table 1 with the text.

R1C6. We appreciate your comments and for this reason, we have revised the manuscript so that all tables and figures are related within the text for a greater clarity of the reader.

R1. Page 6, line 131: change "between" to "among"

R1C7. We want to thank the reviewer for this comment.

R1. Page 6, line 144: change "between" to "among"

R1C8. We want to thank the reviewer for this comment.

R1. Page 6, line 155: change "between" to "among"

R1C9. We want to thank the reviewer for this comment.

R1. Page 7, line 185: change "between" to "among"

R1C10. We want to thank the reviewer for this comment.

R1. Page 8, line 205: change "between" to "among"

R1C11. We want to thank the reviewer for this comment.

R1. Page 9, line 224: to connect Table 2 with the text

R1C12. We want to thank the reviewer for this comment.

R1 Page 13, line 228: I suggest to start the Discussion with the findings related to hypothesis..For example..As it was hypothesized ..

R1C13. As the reviewer suggests, we have connected the start of the Discussion with the experimental hypothesis: “As it was hypothesized, we found different responses to stress, health practices, and self-care according to country, age and gender”.   

R1. Page 13, line 228: I suggest to use additional references to improve the Discussion. Moreover, these references could be also used in the Introduction section. Some suggestions are...

R1C14. We really appreciate this comment and the references suggested which have been included in the discussion and also in the introduction section to tackle more in depth these aspects.

R1. Page 13, line 240: change "between" to "among"

R1.C15. We want to thank the reviewer for this comment.

R1. Page 14, line 314: Although the official language of the four countries is Spanish, it is possible to find some differences in the language. I suggest to include something about this in the limitations.

R1.C16. We really appreciate this comment, this has been included in the limitations.

“and also these instruments were only in the spanish language so it may not capture the full language diversity of each participant country. However, questions related to sociodemographic data, self-care and health activities sections, were adapted by 8 judges (two for each country) to standardize into Spanish from each of the participating countries”.

R1.Page 14, line 322: To add a paragraph with the strength and applications of the findings described in the current study.

R1C17. This implications have been included in the rewritten section of conclusion.

”Due to the unquestionable importance of promoting healthy behaviors and self-care in general, especially during a lockdown, socio-economic, age and gender differences should be considered when addressing health recommendations. Developing strategies to get the most vulnerable groups involved in health behaviors, as well as reinforcing those that serve as a role model, could increase the success of people´s adherence to health guidelines and self-care. This would lead to improving the health, wellness, and well-being of individuals, thus reducing the high costs of medical services.

Therefore, it is mandatory to optimize public policies to make them more health promoting, taking into account the social determinants of health and by altering social norms so that the health of all members and groups of society is a priority. In order to do so, some implications for research, policy and practice are described below.

A more integrative health promotion approach

On one hand, in the same vein that research has noticeable increased on the impact of the COVID-19 pandemic on mental health outcomes, more systematic research is needed to understand the relationship between health behaviours and mental health outcomes to better understand the short- and long-term consequences of this mental health crisis and explore more comprehensives approaches to address it [45, 46]. On the other hand, health policies, measures and media are needed to promote greater health behaviours with a special emphasis on health consciousness and self-care. Enabling people to increase control over their health and its determinants is at the core of health promotion, which paradoxically is more important in this time of crisis than ever before [44]. This health promotion approach can contribute at different levels [47] the downstream level focusing on individual behaviour change and disease management, at the midstream level through interventions affecting organizations and communities and at the upstream level through informing policies affecting the population. As Van den Broucke [44] as pointed out, the expertise with regard to health behaviour change is one of the core competencies of health educators and promoters, and their advice may help governments to achieve the required behaviour change.

A gender-based approach

Since the impact of COVID-19 pandemic and physical distancing measures on health behaviours drive important health inequalities, especially in those disadvantaged groups, further studies should monitor the differential impacts of the current pandemic across age, gender, socioeconomic disadvantage (in early and adult life) and culture/ethnicity and their possible implications to population health and the widening of health inequalities [16]. But also, in order to reduce these health inequities and address immediate and long-term consequences, it is urgent to establish strategies for public health emergencies that take a gender-based approach into account [48,49].

Intersectionality and community action

While preventing the further spread of COVID-19 relies heavily on informing and encouraging the population to adopt protective behaviours, these efforts may be more successful if the advice from experts is combined with local community knowledge [44] and intersectoral strategies [49]. According to The Ottawa Charter, it is crucial in health promotion strategies to emphasize the importance of community action, in the sense of needs assessments, setting priorities, joint planning, capacity building, strengthening local partnerships, intersectoral working and enhancing public participation and social support [50]. All of these activities are aimed to create empowered communities, where individuals and organizations apply their skills and resources in collective efforts to address health priorities and meet their respective health needs. Therefore, community engagement can make a substantial difference in health outcomes, and strengthen the capacity to deal with the negative consequences of the pandemic at individual, organizational and community level.”

R1.Page 14, line 322: I suggest to add in the Conclusion some general information about findings among the four countries.

R1C18. We really appreciate this suggestion and have now included this information in the conclusion section (see R1C17).

Cordially

Reviewer 2 Report

This is a very interesting paper on the field of the impact of COVID-19 first outbreak on stress, health practices, self-care activities in Hispanic countries. This is a large and representative sample, so I consider that it would be a good paper to be published in this journal.

One of the most important strenghts of the paper is that the authors present and interpret their findings according to two important variables: Age and gender. The most important result and conclusion is that future studies should address specific approaches to this situation according to both these variables.

However, prior to the publication, I consider that some minor changes should be addressed.

As the authors made a stratified analysis, I would consider to add some more information in the introduction section. That means, if age and  gender are both variables to take into account when revising results, I think it is very important to revise studies or investigations showing the impact of age and gender on responses to stress, health practices and self-care. Initially, I would add some references for a general topic of stress; and in a second step, I think it should be added gender differences in the approach according to age and gender.

In the methods section, the authors have described the Perceived Stress Scale as a potential scale to evaluate stress in this pandemic situation. I think that this scale should be further described. Has it been used in other studies evaluating stress in COVID pandemic? in general population as well as in professionals attending these patients? The answer to these questions should be mentioned in this paragraph (lines 77-82).

Procedures for statistical analyses are well described and clear for the readers. Gender and age were included as independent variables. There no comment to improve this part.

Figures and Tables reflect in a good manner what they found in this study.

In the discussion section, I consider that the authors emphasized the role of age on adherence to health guidelines, etc, but the discussion of results according to gender should be improved.

I would add some more lines and 2-3 references about potential gender differences in self-care abilities, perceived control and self-care behaviors in the general population. Furthermore, there are several studies focusing on gender differences in health promotion and prevention of stress in populations with mental disorders. Adding a paragraph and 2-3 more references would be good.

The conclusions section is brief. I think it should be expanded with 2-3 more sentences, particularly focusing on the relevance of age and gender in health recommendations.

Author Response

Dear Reviewer 

We would like to thank the editor and the referees for their expeditious and professional review and helpful comments. We answered all their questions and modified the manuscript according to their suggestions. The reviewers' advice have led us to improve the revised version of the manuscript. We have highlighted in yellow the paragraphs that imply changes in the new version of the manuscript.

This is a very interesting paper on the field of the impact of COVID-19 first outbreak on stress, health practices, self-care activities in Hispanic countries. This is a large and representative sample, so I consider that it would be a good paper to be published in this journal.

One of the most important strengths of the paper is that the authors present and interpret their findings according to two important variables: Age and gender. The most important result and conclusion is that future studies should address specific approaches to this situation according to both these variables.

However, prior to the publication, I consider that some minor changes should be addressed.

R2. As the authors made a stratified analysis, I would consider to add some more information in the introduction section. That means, if age and  gender are both variables to take into account when revising results, I think it is very important to revise studies or investigations showing the impact of age and gender on responses to stress, health practices and self-care. Initially, I would add some references for a general topic of stress; and in a second step, I think it should be added gender differences in the approach according to age and gender.

R2C1.We thank the reviewer for this suggestion that has improved the rationale of the study. We have add this information and relevant recent evidence in the field in the introduction

“In the same vein, people who suffered a higher impact of quarantine on their physical ac-tivities tend to have higher prevalence of anxiety and depression symptoms  and keeping active can play a very important protective factor of mental and physical health [14,15].

However, stress responses, self-care or health behaviour changes during COVID-19 lockdown can differ by age, gender, ethnicity, and socioeconomic position. This situation seems particularly stressful for younger adults (< 35 years), women, people without work and with low incomes [16]. Moreover ethnic minorities showed to undertake less exercise and consume lower amounts of fruit and vegetables during lockdown. Regarding coun-tries differences, it has been found that living in a high‐income country during the pan-demic is a risk factor for depression and anxiet and countries belonging to the Latin America and Caribbean cluster showed a lower prevalence of mental health symptoms compared to countries belonging to North America, Europe and Central Asia, and Sub-Saharan Africa clusters [17].”

R2. In the methods section, the authors have described the Perceived Stress Scale as a potential scale to evaluate stress in this pandemic situation. I think that this scale should be further described. Has it been used in other studies evaluating stress in COVID pandemic? in general population as well as in professionals attending these patients? The answer to these questions should be mentioned in this paragraph (lines 77-82).

R2C2.We thank the reviewer for this suggestion. We have add other studies evaluating stress in COVID pandemic general population as well as in professionals using PSS-10.

“Studies have been published that report in relation to PSS-10, optimal psychometric prop-erties, both in the general population and people exposed to confinement [19,20] and specifically in health professionals who attend the emergency situation [21]”.

R2. Procedures for statistical analyses are well described and clear for the readers. Gender and age were included as independent variables. There no comment to improve this part.

R2C3.We want to thank the reviewer for this comment.

R2. Figures and Tables reflect in a good manner what they found in this study.

R2C4.We want to thank the reviewer for this comment.

R2 In the discussion section, I consider that the authors emphasized the role of age on adherence to health guidelines, etc, but the discussion of results according to gender should be improved.I would add some more lines and 2-3 references about potential gender differences in self-care abilities, perceived control and self-care behaviors in the general population. Furthermore, there are several studies focusing on gender differences in health promotion and prevention of stress in populations with mental disorders. Adding a paragraph and 2-3 more references would be good.

R2C5. We have improved this part of the discussion according the reviewer suggestions.

Concerning gender, the fact that women are reporting higher scores on perceived stress was an expected outcome consistent with previous studies during lockdown [23]. But this differential impact of gender on mental health outcomes goes beyond the pan-demic situation, and could be argued to stem from women’s roles, mainly the burden of both work and caring roles, as well as to the number of social roles women fulfil [31].These gender differences may also contribute to a greater vulnerability for women, not only by contracting the virus, given that the majority of workers in frontline sectors are women, but also because of an overload of their caring role and therefore, an experience of greater stress levels and mental health problems [34].

R2. The conclusions section is brief. I think it should be expanded with 2-3 more sentences, particularly focusing on the relevance of age and gender in health recommendations.

R2C6.  In line with the all reviewers suggestions, the conclusion section has been rewritten and amended (see R1C17;R1C18; R2C6;R4C5).

“Due to the unquestionable importance of promoting healthy behaviors and self-care in general, especially during a lockdown, socio-economic, age and gender differences should be considered when addressing health recommendations. Developing strategies to get the most vulnerable groups involved in health behaviors, as well as reinforcing those that serve as a role model, could increase the success of people´s adherence to health guidelines and self-care. This would lead to improving the health, wellness, and well-being of individuals, thus reducing the high costs of medical services.

Therefore, it is mandatory to optimize public policies to make them more health promoting, taking into account the social determinants of health and by altering social norms so that the health of all members and groups of society is a priority. In order to do so, some implications for research, policy and practice are described below.

A more integrative health promotion approach

On one hand, in the same vein that research has noticeable increased on the impact of the COVID-19 pandemic on mental health outcomes, more systematic research is needed to understand the relationship between health behaviours and mental health outcomes to better understand the short- and long-term consequences of this mental health crisis and explore more comprehensives approaches to address it [45, 46]. On the other hand, health policies, measures and media are needed to promote greater health behaviours with a special emphasis on health consciousness and self-care. Enabling people to increase control over their health and its determinants is at the core of health promotion, which paradoxically is more important in this time of crisis than ever before [44]. This health promotion approach can contribute at different levels [47] the downstream level focusing on individual behaviour change and disease management, at the midstream level through interventions affecting organizations and communities and at the upstream level through informing policies affecting the population. As Van den Broucke [44] as pointed out, the expertise with regard to health behaviour change is one of the core competencies of health educators and promoters, and their advice may help governments to achieve the required behaviour change.

A gender-based approach

Since the impact of COVID-19 pandemic and physical distancing measures on health behaviours drive important health inequalities, especially in those disadvantaged groups, further studies should monitor the differential impacts of the current pandemic across age, gender, socioeconomic disadvantage (in early and adult life) and culture/ethnicity and their possible implications to population health and the widening of health inequalities [16]. But also, in order to reduce these health inequities and address immediate and long-term consequences, it is urgent to establish strategies for public health emergencies that take a gender-based approach into account [48,49].

Intersectionality and community action

While preventing the further spread of COVID-19 relies heavily on informing and encouraging the population to adopt protective behaviours, these efforts may be more successful if the advice from experts is combined with local community knowledge [44] and intersectoral strategies [49]. According to The Ottawa Charter, it is crucial in health promotion strategies to emphasize the importance of community action, in the sense of needs assessments, setting priorities, joint planning, capacity building, strengthening local partnerships, intersectoral working and enhancing public participation and social support [50]. All of these activities are aimed to create empowered communities, where individuals and organizations apply their skills and resources in collective efforts to address health priorities and meet their respective health needs. Therefore, community engagement can make a substantial difference in health outcomes, and strengthen the capacity to deal with the negative consequences of the pandemic at individual, organizational and community level.”

Cordially

Reviewer 3 Report

Thank you for the opportunity to review this paper. The study involved the use of an online survey that included items from the PSS-10, SASS-14, and questions about COVID-19 health practices developed by the authors. The survey was posted on social media and 1,082 respondents from three different Hispanic countries were included in the analysis. The authors performed a descriptive statistics analysis, MANOVA, and ANCOVA to learn how country, gender, and age were related to healthy behaviors. Overall, the paper is well written and there is much to like. The results of the study indicated that young women suffered greater levels of stress and demonstrated a lower level of adherence to health guidelines in all three countries. This seems to be an important finding in the larger discussion of what public health strategies to adopt in response to the pandemic.

The discussion section stated that differences between countries may be due to differences in the number of days people were confined. Additional details would be helpful here. What is the difference exactly?

I commend the authors for their work on this project.

Author Response

Dear Reviewer

We would like to thank the editor and the referees for their expeditious and professional review and helpful comments. We answered all their questions and modified the manuscript according to their suggestions. The reviewers' advice have led us to improve the revised version of the manuscript. We have highlighted in yellow the paragraphs that imply changes in the new version of the manuscript.

Thank you for the opportunity to review this paper. The study involved the use of an online survey that included items from the PSS-10, SASS-14, and questions about COVID-19 health practices developed by the authors. The survey was posted on social media and 1,082 respondents from three different Hispanic countries were included in the analysis. The authors performed a descriptive statistics analysis, MANOVA, and ANCOVA to learn how country, gender, and age were related to healthy behaviors. Overall, the paper is well written and there is much to like. The results of the study indicated that young women suffered greater levels of stress and demonstrated a lower level of adherence to health guidelines in all three countries. This seems to be an important finding in the larger discussion of what public health strategies to adopt in response to the pandemic.

R3 The discussion section stated that differences between countries may be due to differences in the number of days people were confined. Additional details would be helpful here. What is the difference exactly?

R3C1. During data collection, we verified that the virus had a different impact on each country. This fact can be seen in the following figure, which shows that after the start of the pandemic in China, the number of positives caused by the virus began to increase in Spain after 35 days, in Chile and Ecuador approximately at 40 days, and in Colombia approximately 50 days after the virus began to hit China. The dotted red lines indicate the start and end dates of data collection, and show that in this time window, the impact of covid-19 on each country (measured from the number of positives) was different, so Spain had the greatest impact, followed by Chile and Ecuador, and finally Colombia.

Total number of positive cases by day and country (i.e., in log form) since the start of the pandemy in China (starting date: 22-01-2020). CHN: China, SPN: Spain, COL: Colombia, CHL: Chile, ECU: Ecuador

Data-series source: John Hopkins University (data available from https://coronavirus.jhu.edu/map.html, Dong et al., 2020).

R3. I commend the authors for their work on this project.

R3.C2. We want to thank the reviewer for this comment.

Cordially

Reviewer 4 Report

This study presents some very interesting results that will help in the development of effective public health strategies.

In general it seems to me a good study on the methodology used and the results obtained. However, I have small comments that I hope will help to improve some aspects.

In the introduction, lines 36-38, Data are shown on the deaths that have occurred due to covid-19 in the different countries analyzed, but it is not clear to me the origin of the source, if it is associated with those that appear at the end or in this case they are not mentioned.

At the end of the introduction, it is recommended to include a paragraph briefly explaining the blocks that make up the investigation and which are detailed below.

In the 3.3.2 Adherence ti Public Health guidelines, line 181, the word “adherence” appears divided between lines 181 and 182, separating two consonants, perhaps the whole word should be lowered because two consonants of the same syllable are separated.

In the Discussion, lines 249-250, it is stated that the disease in South America has not been as severe as in Spain, perhaps the source of this information should be included.

Finally, the conclusions are too short, it is recommended to also incorporate the main contribution of this study to science and practical implications.

Author Response

Dear Reviewer 

We would like to thank the editor and the referees for their expeditious and professional review and helpful comments. We answered all their questions and modified the manuscript according to their suggestions. The reviewers' advice have led us to improve the revised version of the manuscript. We have highlighted in yellow the paragraphs that imply changes in the new version of the manuscript.

Comments and Suggestions for Authors

This study presents some very interesting results that will help in the development of effective public health strategies.

In general it seems to me a good study on the methodology used and the results obtained. However, I have small comments that I hope will help to improve some aspects.

R4. In the introduction, lines 36-38, Data are shown on the deaths that have occurred due to covid-19 in the different countries analyzed, but it is not clear to me the origin of the source, if it is associated with those that appear at the end or in this case they are not mentioned.

R4C1.We thank the reviewer for his suggestion. We have clarified in the document the source of the data: “Data-series source: John Hopkins University (data available from https://coronavirus.jhu.edu/map.html, Dong et al., 2020). Regarding the statement made in the introduction about the number of deaths we wanted to reflect the different impact that the pandemic has caused in each country. As can be seen in the graph and table shown below, the impact of covid, measured from the logarithm of the number of deaths during the data collection period (from March 31 to April 14, indicated with dotted red lines) was different in each country. Despite the fact that we have not used the data on the number of deaths at the time of data collection in our analyzes, we do collect the number of days of confinement of the population as a possible measure of stress, although we have not found a correlation with the response variables.

Total number of deaths by day and country (i.e., in log form) since the start of the pandemy in China (starting date: 22-01-2020). CHN: China, SPN: Spain, COL: Colombia, CHL: Chile, ECU: Ecuador

China

Spain

Ecuador

Colombia

Chile

31.03.2020

3308

8464

75

16

12

14.04.2020

3345

18056

369

127

92

We have updated the statistics in the introduction according to the Pandemic context of the specific countries during data acquisition (i.e., from 31st of March to 14th of April). We attach the total number of deaths by date and country for the last day of the acquisition (14th of April).

“As of 14st April 2020, more than 18056 people had died as a result of the coronavirus disease 2019 (COVID-19) in Spain; more than 127 in Colombia; more than 92 in Chile; and more than 369 in Ecuador [1]”

R4. At the end of the introduction, it is recommended to include a paragraph briefly explaining the blocks that make up the investigation and which are detailed below.

R4C2. As the reviewer has suggested, a paragraph has been included at the end of the introduction explaining the importance of the blocks that the study is based on: gender, age and country groups (related to R1C2).

“Therefore, in this study we hypothesized the existence of differences in responses to stress, health practices and self-care activities depending on country, age and gender due to mandatory COVID-19 confinement”.

R4. In the 3.3.2 Adherence to Public Health guidelines, line 181, the word “adherence” appears divided between lines 181 and 182, separating two consonants, perhaps the whole word should be lowered because two consonants of the same syllable are separated.

R4C3. We want to thank the reviewer for this comment and this was assumed in the manuscript

R4. In the Discussion, lines 249-250, it is stated that the disease in South America has not been as severe as in Spain, perhaps the source of this information should be included.

R4C4. We have attached the source: “It may be because of the impact of the coronavirus disease on South America was not as severe as it was in Spain at that point according to epidemiological data from Johns Hopkins University website [1]. In addition, Chile reported a lower percentage of males in charge of children and older people, which may also be a factor in the lower levels of stress suffered by this group, thus enabling them to maintain healthier daily routines and report better adherence to health guidelines [2] “.

R4. Finally, the conclusions are too short, it is recommended to also incorporate the main contribution of this study to science and practical implications.

R4C5. In the same way that in R1C17, R1C18 and R2C6 comments,  we have improved this section thanks to reviewers’ recommendations.

“Due to the unquestionable importance of promoting healthy behaviors and self-care in general, especially during a lockdown, socio-economic, age and gender differences should be considered when addressing health recommendations. Developing strategies to get the most vulnerable groups involved in health behaviors, as well as reinforcing those that serve as a role model, could increase the success of people´s adherence to health guidelines and self-care. This would lead to improving the health, wellness, and well-being of individuals, thus reducing the high costs of medical services.

Therefore, it is mandatory to optimize public policies to make them more health promoting, taking into account the social determinants of health and by altering social norms so that the health of all members and groups of society is a priority. In order to do so, some implications for research, policy and practice are described below.

A more integrative health promotion approach

On one hand, in the same vein that research has noticeable increased on the impact of the COVID-19 pandemic on mental health outcomes, more systematic research is needed to understand the relationship between health behaviours and mental health outcomes to better understand the short- and long-term consequences of this mental health crisis and explore more comprehensives approaches to address it [45, 46]. On the other hand, health policies, measures and media are needed to promote greater health behaviours with a special emphasis on health consciousness and self-care. Enabling people to increase control over their health and its determinants is at the core of health promotion, which paradoxically is more important in this time of crisis than ever before [44]. This health promotion approach can contribute at different levels [47] the downstream level focusing on individual behaviour change and disease management, at the midstream level through interventions affecting organizations and communities and at the upstream level through informing policies affecting the population. As Van den Broucke [44] as pointed out, the expertise with regard to health behaviour change is one of the core competencies of health educators and promoters, and their advice may help governments to achieve the required behaviour change.

A gender-based approach

Since the impact of COVID-19 pandemic and physical distancing measures on health behaviours drive important health inequalities, especially in those disadvantaged groups, further studies should monitor the differential impacts of the current pandemic across age, gender, socioeconomic disadvantage (in early and adult life) and culture/ethnicity and their possible implications to population health and the widening of health inequalities [16]. But also, in order to reduce these health inequities and address immediate and long-term consequences, it is urgent to establish strategies for public health emergencies that take a gender-based approach into account [48,49].

Intersectionality and community action

While preventing the further spread of COVID-19 relies heavily on informing and encouraging the population to adopt protective behaviours, these efforts may be more successful if the advice from experts is combined with local community knowledge [44] and intersectoral strategies [49]. According to The Ottawa Charter, it is crucial in health promotion strategies to emphasize the importance of community action, in the sense of needs assessments, setting priorities, joint planning, capacity building, strengthening local partnerships, intersectoral working and enhancing public participation and social support [50]. All of these activities are aimed to create empowered communities, where individuals and organizations apply their skills and resources in collective efforts to address health priorities and meet their respective health needs. Therefore, community engagement can make a substantial difference in health outcomes, and strengthen the capacity to deal with the negative consequences of the pandemic at individual, organizational and community level.”

Cordially

Round 2

Reviewer 1 Report

Congratulations. The manuscript was improved and I recommend its acceptance.